# StairwayToStain: A Gradual Stain Translation Approach for Glomeruli Segmentation

**Ali Alhaj Abdo**                                    ALI.AL-HAJ-ABDO@ETU.UNISTRA.FR

**Islem Mhiri**                                                  I.MHIRI@UNISTRA.FR

**Zeeshan Nisar**                                                 NISAR@UNISTRA.FR
*ICube, University of Strasbourg, Illkirch-Graffenstaden 67400, France.*

**Barbara Seeliger**                      BARBARA.SEELIGER@IHU-STRASBOURG.EU
*IHU, Strasbourg 67000, France.*

**Thomas Lampert**                                            LAMPERT@UNISTRA.FR
*ICube, University of Strasbourg, Illkirch-Graffenstaden 67400, France.*

## Abstract

Image-to-image translation (I2I) has advanced digital pathology by enabling knowledge transfer across clinical contexts through unsupervised domain adaptation (UDA). Although promising, most I2I frameworks transfer source-labeled data to target unlabeled data directly in a one-off way. However, translating stains from information-poor domains to information-rich ones can lead to a domain shift problem due to the large discrepancy between domains. To address this issue, we propose StairwayToStain (STS), an unsupervised gradual stain translation framework that uses intermediate stains to bridge the gap between the source and target stain. Our method is grounded in three main phases: (i) measuring the domain shift between different stains, (ii) defining a translation path, and (iii) performing the gradual stain translation. Our method demonstrates its efficacy in improving glomeruli segmentation when translating from immunohistochemical (IHC) to histochemical stains, as well as between different IHC stains. Comprehensive experiments on stain translation demonstrate STS's competitive results compared to its variants and state-of-the-art direct I2I methods in achieving UDA. Moreover, we are able to generate additional stains during the translation process. Our method presents the first framework for gradual domain adaptation in stain translation.

**Keywords:** Digital Pathology, Stain Translation, Domain Adaptation, Domain Shift

## 1 Introduction

Traditional histopathological analysis is characterised by its time-consuming nature, limited accessibility, and variability. This analysis is routinely conducted through microscopic examination, using various stains to highlight different features. As scanning technology advanced in the past few decades, a new paradigm shift has been introduced to address these challenges. Building on these advances, a new field emerged: digital pathology. This field brings many advantages to pathologists, one being the digitisation of slides into whole slide images (WSIs). Specifically, this offers streamlined workflows, enabling rapid image acquisition, storage, and analysis while ensuring non-destructive examination of specimens.

Whole slide images created a surge in histopathological data, allowing for the development of Computer-Aided Diagnosis (CAD) systems and machine learning (ML) models,

which form the field of computational pathology. Although promising, deploying such systems into clinical settings faces significant challenges. Usually, these models are trained on a certain stain that follows a specific protocol. However, different staining and scanning protocols are used in different medical facilities, which leads to inter- and intra-stain variabilities in the distribution of images, resulting in a phenomenon commonly referred to in ML as *domain shift*. This leads to a failure in generalising ML models to different stains and staining protocols (Nisar et al., 2022). Theoretically, one can create another dataset through staining and annotation to retrain a network. However, this methodology is time-consuming and requires a lot of human resources. Therefore, it is impractical to perform this for every required stain or preparation protocol variation. Most domain shift research in histopathology has focused on addressing intra-stain variations to enhance model robustness within a specific stain (Khan et al., 2014; Zanjani et al., 2018). In contrast, inter-stain variations have received limited attention.

With the increasing success of generative adversarial networks (GANs) (Goodfellow et al., 2014), image-to-image translation (I2I), or stain transfer, methods are being used to perform unsupervised domain adaptation (UDA), either to reduce domain shift (Gadermayr et al., 2018) or when training stain invariant models through data augmentation (Vasiljević et al., 2021b, 2023). Notably, Gadermayr et al.'s Multi-Domain Supervised 1 (MDS1) (2019) approach achieves UDA using GAN-based image translation to translate unlabeled target domain images to resemble source domain images, for which a pre-trained supervised model is available. Zhu et al.'s CycleGAN (2017) is a prominent example of a GAN-based image translation architecture that is frequently used. While effective (Zingman et al., 2024), it can only perform stain transfer between two stains at a time. Thus, many networks need to be trained to obtain multiple stains. StarGAN (Choi et al., 2018) is a multi-domain transfer network, which can be used for stain transfer, but its results are subpar when compared to the CycleGAN (Vasiljević et al., 2021b). Currently, Vasiljević et al.'s HistoStarGAN (2023) holds the state-of-the-art (SOTA) results for stain-invariant segmentation. It is also the first model that is capable of simultaneously performing stain transfer, stain normalisation, and stain invariant segmentation. This model has also shown capabilities of generalising to multiple unseen stains without needing any supplementary annotations. However, some artefacts can be present in the translations.

While these stain transfer solutions are usually effective, they still struggle to mitigate domain shift when translating from information-poor to information-rich domains (Vasiljević et al., 2021b). In histopathology, information-poor domains refer to those that highlight or mark limited biological features, such as immunohistochemical stains (IHC). On the other hand, information-rich domains are characterised by a broader representation of biological features, such as histochemical stains. For such stain transfers, it has been shown that the translation network must include imperceptible noise during the translation process to fulfill cycle consistency (Vasiljević et al., 2021a). Such noise is essential to accurately reconstruct the specific IHC markers in their correct positions, which can otherwise not be implied from general histochemical stains, however, it hinders the performance of downstream tasks such as MDS1 segmentation because it induces a domain shift.

This work aims to reduce the amount of noise present in translated images to increase segmentation accuracy. Instead of directly translating from IHC to histochemical stains, or between different IHC stains, which have large information gaps, we propose Stairway-

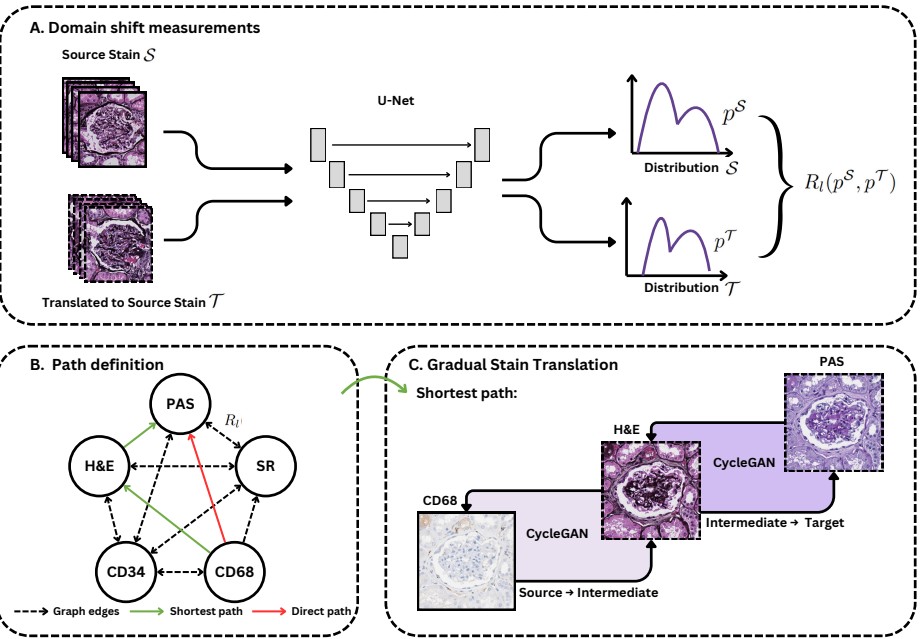

Figure 1: Overview of StairwayToStain, our gradual stain translation framework. (A) The domain shift metric, $R_l(p^{\mathcal{S}}, p^{\mathcal{T}})$ (Stacke et al., 2021), is measured by extracting features from source stain $\mathcal{S}$ and target translated to source stain $\mathcal{T}$ using a source stain pre-trained U-Net. (B) The domain shift scores extracted are used to build a graph using the stains as nodes. A path finding algorithm can then be used to find the shortest translation path in terms of domain shift. (C) The translation is performed using CycleGANs between the ordered stains.

ToStain, an unsupervised technique that uses intermediate stains to perform gradual translation. In this way, the information gap between intermediate stains is minimised, which reduces the amount of noise necessary to achieve the translation. Moreover, Wang et al. (2022) have shown that gradual translation between domains with lower domain shifts can lead to more accurate translation. We first measure the distance between different stains using the domain shift metric (DSM) (Stacke et al., 2021). Then, to define the proximity of stain modalities, we use Dijkstra's path-finding algorithm. This helps us, order the intermediate stains and then CycleGAN is used to perform the translations between the stains.

This approach is greatly inspired by the merge sort technique, where we breakdown a long non-linear task into smaller tasks that are easier to solve. To the best of our knowledge, this represents the first gradual domain adaptation framework for stain translation.

## 2 Proposed Method

This section presents the key steps of StairwayToStain, as outlined in Figure 1.

## 2.1 Domain Shift Measurements

To effectively perform gradual stain translation, a metric is needed to measure domain shift. For this, we choose the domain shift metric with Wasserstein distance (Stacke et al., 2021), which uses the feature representations of a pre-trained model to measure the difference between the distributions of a source domain and another domain translated to it, as seen in Figure 1A. By measuring the difference between the source stain and the target translated to the source stain, we obtain the domain shift score (DSS) which allows us to estimate the difficulty of translation between the stain pairs in a directional manner. Nisar et al. (2022) have shown that this metric has a strong correlation with the segmentation performance in stain-translated data and concluded that it can be used to predict the performance of pre-trained neural networks on unseen stains.

To formalise the DSM, consider a convolutional neural network with layers $l_1, \ldots, l_L$. Activations at layer $l$ and filter $k$ are denoted as $\Phi_l(x) = \phi_{l1}(x), \ldots, \phi_{lk}(x)$, where $\phi_{lk}(x) \in \mathbb{R}^{hxw}$. The mean of each $\phi_{lk}(x)$ is given by $c_{lk}(x) = \frac{1}{hw} \sum_{i,j}^{h,w} \Phi_{lk}(x)_{i,j}$. Let $p_{c_{lk}}^{\mathcal{S}}$ denote the continuous distribution of $c_{lk}(x)$ over the source stain $\mathcal{S}$ and $p_{c_{lk}}^{\mathcal{T}}$ denote the continuous distribution of $c_{lk}(x)$ over the target stain $\mathcal{T}$. If $\pi(p_{c_{lk}}^{\mathcal{S}}, p_{c_{lk}}^{\mathcal{T}})$ represents the joined distribution with margins $p_{c_{lk}}^{\mathcal{S}}$ and $p_{c_{lk}}^{\mathcal{T}}$, the Wasserstein distance $\mathcal{W}$ can be formally defined as

$$\mathcal{W}(p_{c_{lk}}^{\mathcal{S}}, p_{c_{lk}}^{\mathcal{T}}) = \inf_{\pi \in \Gamma(p_{c_{lk}}^{\mathcal{S}}, p_{c_{lk}}^{\mathcal{T}})} \int_{RxR} |(x-y)| d\pi(x,y). \tag{1}$$

This distance depends on the absolute values of the generated distributions. As $\mathcal{S}$ and $\mathcal{T}$ get closer, $\mathcal{W}(p_{c_{lk}}^{\mathcal{S}}, p_{c_{lk}}^{\mathcal{T}}) \to 0$. As such, it measures the shape discrepancy and distance between the two distributions. The domain shift metric (DSM) $R_l$ can now be defined as

$$R_l(p^{\mathcal{S}}, p^{\mathcal{T}}) = \frac{1}{k} \sum_{i=1}^{k} \mathcal{W}(p_{c_{lk}}^{\mathcal{S}}, p_{c_{lk}}^{\mathcal{T}}). \tag{2}$$

## 2.2 Gradual Domain Adaptation Paths

Using the domain shift scores measured between each pair of stains (source & target→source) as edge weights, we build a directional graph in which the nodes represent stains. We consider the GDA path to be a shortest-path problem, i.e. we seek the shortest translation path (in terms of translated domain shift) between two stains using a path-finding algorithm.

In our case, we use Dijkstra's algorithm (1959). This algorithm extracts the shortest possible translation path from a given stain towards another. As such, the algorithm can provide two types of paths. Either it outputs a direct translation path (Source→Target) if that path represents the least domain shift, or a gradual translation path (Source → Intermediate$_{1...n}$→Target), while ensuring that no stain is repeated in the path (i.e. avoiding cyclic translations). An example of this is presented in Figure 1B, in which the direct translation CD68→PAS is replaced with the indirect translation CD68→H&E→PAS.

## 2.3 Gradual Stain Translation

With the paths extracted, we perform stain translation. Instead of directly translating from a source to target stain, we take a path that includes intermediate stains to bridge

the gap between information-poor and information-rich stains. This breaks down the stain translation process into smaller easier-to-solve mappings. To perform these translations we use Zhu et al.'s (2017) original CycleGAN. In this architecture, two generators, $G_{AB}$ and $G_{BA}$, are used to translate from stain $A$ to $B$ and vice versa. Moreover, two discriminators are present, $D_A$ and $D_B$, that discriminate between real and generated images for stains $A$ and $B$ respectively. This network is constrained with the adversarial loss ($\mathcal{L}_{\text{adv}}$), cycle-consistency loss $\mathcal{L}_{\text{cyc}}$, and identity loss $\mathcal{L}_{\text{id}}$. CycleGAN's cycle-consistency loss ensures reversibility between the images during translation, which allows biological structures within tissues to retain their position and morphology. This makes it ideal for UDA, and eventually GDA. The full objective function is $\mathcal{L}_{\text{CycelGAN}} = \mathcal{L}_{\text{adv}} + w_{cyc}\mathcal{L}_{\text{cyc}} + w_{id}\mathcal{L}_{\text{id}}$.

## 3 Experiments and Results

Inspired by Nisar et al.'s (2022) work on measuring domain shift, we use the U-Net architecture (Ronneberger et al., 2015) as our pre-trained model for glomeruli segmentation. Each experiment is repeated 15 times by using three trained CycleGANs and five trained U-Nets to account for random variations. We compare STS to direct UDA stain translation, i.e. test time CycleGAN (Gadermayr et al., 2019) and StarGAN translation (MDS1 and MDS*1 respectively). Since it has been shown that translations from CD68 pose problems, the evaluation will focus on its translation to a histochemical and an IHC stain.

### 3.1 Dataset and Evaluation Metrics

Tissue samples were acquired from a cohort of 10 patients with renal carcinoma who underwent tumor nephrectomy. Tissue sections were extracted as far as possible from the tumors to mainly display physiological renal tissue. The paraffin-embedded samples were sliced into $3\ \mu$ sections and then stained using Ventana Benchmark Ultra, an automated staining tool. Five stains were considered: PAS, Jones Hematoxylin and Eosin (H&E), Sirius Red (SR), and two IHC markers CD34 and CD68, that highlight endothelial cells and macrophages receptively. The tissue samples were then scanned using Aperio AT2 to capture the WSIs at 40 magnification (a resolution of 0.253 $\mu$/pixel). All the glomeruli were then annotated and validated by pathologists using Cytomine (Marée et al., 2016). The dataset was split into 4 training, 2 validation, and 4 test patients. The number of glomeruli in each stained dataset is present in Table 1. For the DSM measurements we use 3 subsets of 1000 randomly sampled patches of size $508 \times 508$ pixels because they contain the glomeruli and some of the surrounding tissue at the level of detail required. To evaluate domain adaptation we choose the task of glomeruli segmentation evaluated using the F1 score, precision, and recall metrics.

### 3.2 Training Strategies

The same training parameters were used for all U-Net models: a batch size of 8, a learning rate of 0.0001, 250 epochs, and the network with the lowest validation loss was kept. All patches were standardised to $[0, 1]$ and normalised using the mean and standard deviation of the labelled training set. The non-tissue background of the WSI was removed by thresh-

| Stain | Train | Validation | Test |
|-------|-------|------------|------|
| PAS | 662 | 588 | 1092 |
| Jones H&E | 624 | 593 | 1043 |
| Sirius Red | 654 | 579 | 1049 |
| CD34 | 568 | 598 | 1019 |
| CD68 | 529 | 524 | 1046 |

Table 1: The number of glomeruli in each staining used in the private dataset.

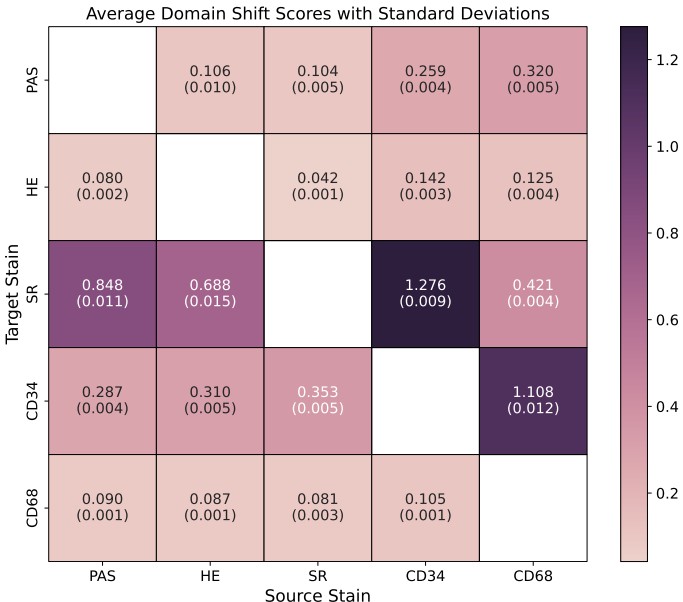

Figure 2: Average Domain Shift Scores ($R_l$) of 3 sets of 1000 randomly sampled patches translated from a source to target stains. Standard deviations are in parentheses.

olding each image by its mean value, then removing small objects and closing holes. We followed Lampert et al.'s (2019) training augmentation strategy and parameters.

Gadermayr et al. (2018) showed that different sampling strategies can negatively impact CycleGAN's performance, we randomly extract patches using a uniform sampling strategy (in an unsupervised manner). The loss weights and architecture were taken from the original CycleGAN paper since they produced realistic images ($w_{cyc} = 10, w_{id} = 5$) (Zhu et al., 2017). The same training strategy as Vasiljević et al. (2021b) was employed.

### 3.3 Results

The extracted DSSs are present in Figure 2. When applying STS on the CD68 stain, the translation path CD68 → H&E → PAS is obtained for the CD68 → PAS translation, and CD68 → H&E → CD34 for the IHC → IHC path. We can also calculate a cumulative STS domain shift score, as the sum of the individual translations. For example, for the CD68

| Translation | Direct DSS | Cumulative STS DSS | Measured STS DSS |
|---|---|---|---|
| CD68 → PAS | 0.320 | 0.231 | 0.260 |
| CD68 → CD34 | 1.108 | 0.435 | 0.387 |

Table 2: Different domain shift scores for each translation path.

→ PAS, a cumulative STS DSS of 0.231 (0.125 + 0.106) is extracted, which is lower than the direct translation DSS (0.320). Moreover, we remeasure the domain shift, referred to as measured STS DSS, when using the STS gradual translation and obtain reduced DSSs for each of our chosen translation paths, further confirming the reduction of domain shift with our method. The direct, cumulative STS, and measured STS are present in Table 2. By following these paths, we transform the CD68 images towards their target stains, as shown in Figure 1C for CD68 → PAS.

We can then test the segmentation performance of segmentation U-Nets (pre-trained on the target stains) on the translated images using the CD68 labels. The results are presented in Table 3. StairwayToStain outperforms both direct approaches using the CycleGAN and the StarGAN, obtaining an overall higher F1 score.

### 3.4 Discussion

We have shown that our STS framework can reduce domain shift when compared to direct stain translation in the five stains chosen in this study. This leads to better glomeruli segmentation, particularly when translating from the CD68 towards PAS, and CD68 towards CD34. This shows that a gradual translation approach indeed leads to better segmentation performance when using MDS1 in the case where the domain shift is particularly large.

This improvement is observed for F1 and precision, but there is a reduction in recall for PAS and it is equaled for CD34. Nevertheless, the F1 score in the case of CD34 is still far below an acceptable level (even though the translations are visually plausible, see the supplementary material for examples). The reason for these observations is still being investigated but we hypothesise that noise remains in the translated images, although perhaps in a different form that effects the segmentation model differently. Eventually, more advanced models should be developed to take this into account. Moreover, this method can be further extended to integrate more, and more similar, stains into the framework, allowing the formation of shorter and/or more fine grained translation paths between stains.

We now compare these results to those of current SOTA stain-invariant segmentation models such as HistoStarGAN, which has an F1 score of 0.755 (0.006), a precision of 0.845 (0.039), and a recall of 0.684 (0.024) in the same dataset for CD68. This high performance can be attributed to the fact that the model is trained to be stain-invariant through stain augmentation, and is therefore exposed to several stains during training (as such, it has the limitation of requiring access to all stains during training, whereas MDS1 can be adapted as necessary by training additional translation networks). Nevertheless, the stain augmentation used in HistoStarGAN (as well as another stain invariant approach called UDAGAN (Vasiljević et al., 2021b)) is performed by (direct) stain translation and therefore also includes the imperceptible noise. It could therefore be replaced with the proposed gradual strain translation, which should lead to an increase in their performance.

| Method | Score | Target Stain | |
|---|---|---|---|
| | | PAS | CD34 |
| vStain | F1 score | 0.001 (0.002) | 0.006 (0.009) |
| | Precision | 0.097 (0.129) | 0.005 (0.004) |
| | Recall | 0.001 (0.001) | 0.027 (0.042) |
| MDS1 (Gadermayr et al., 2019) | F1 score | 0.608 (0.035) | 0.165 (0.032) |
| | Precision | 0.747 (0.068) | 0.328 (0.031) |
| | Recall | **0.521 (0.061)** | 0.112 (0.026) |
| MDS*1 (Choi et al., 2018) | F1 score | 0.092 (0.055) | 0.002 (0.003) |
| | Precision | 0.242 (0.116) | 0.095 (0.068) |
| | Recall | 0.061 (0.044) | 0.001 (0.001) |
| StairwayToStain (STS) | F1 score | **0.617 (0.044)** | **0.195 (0.045)** |
| | Precision | **0.849 (0.039)** | **0.606 (0.033)** |
| | Recall | 0.488 (0.060) | **0.118 (0.034)** |

Table 3: Quantitative results for domain adaptation from CD68 towards target stains applied using different methods. vStain refers to applying a stain pre-trained U-Net directly on CD68 images. The highest scores are highlighted in bold.

## 4 Conclusion

This article has presented StairwayToStain, the first approach to GDA for stain translation. First, we used the domain shift metric to measure the distance between different stains. Second, we used Dijkstra's algorithm to find paths between different stains to apply a gradual stain transfer approach, instead of direct translation. Then, we employed Cycle-GANs for translation. Our method outperforms the most commonly used direct translation approaches. This work serves as a proof of concept to the methodology. To further improve on this approach, a dedicated network will need to be developed, removing the need for multiple networks. Another direction would be to integrate this methodology into stain-invariant augmentation strategies, used by, for example, HistoStarGAN and UDAGAN.

## Acknowledgments and Disclosure of Funding

This work of the Interdisciplinary Thematic Institute HealthTech, part of the ITI 2021-2028 program of the University of Strasbourg, CNRS and Inserm, was supported by IdEx Unistra (ANR10-IDEX-0002) and SFRI (STRAT'US project, ANR-20-SFRI-0012) under the framework of the French Investments for the Future Program. It was supported by ANR HistoGraph (ANR-23-CE45-0038). We acknowledge the ERACoSysMed & e:Med initiatives by BMBF, SysMIFTA (managed by PTJ, FKZ 031L-0085A; ANR ANR-15-CMED-0004), Prof. Cédric Wemmert, and Prof. Friedrich Feuerhake and MHH for the high-quality images & annotations: N. Kroenke, N. Schaadt, V. Volk & J. Schmitz. We thank Nvidia, the *Centre de Calcul* (University of Strasbourg) & GENCI-IDRIS (2020-A0091011872) for GPU access. The authors have no competing interests that are relevant to this article's content.

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

## Supplementary Material

This section contains additional images of translations paths and translated patches.

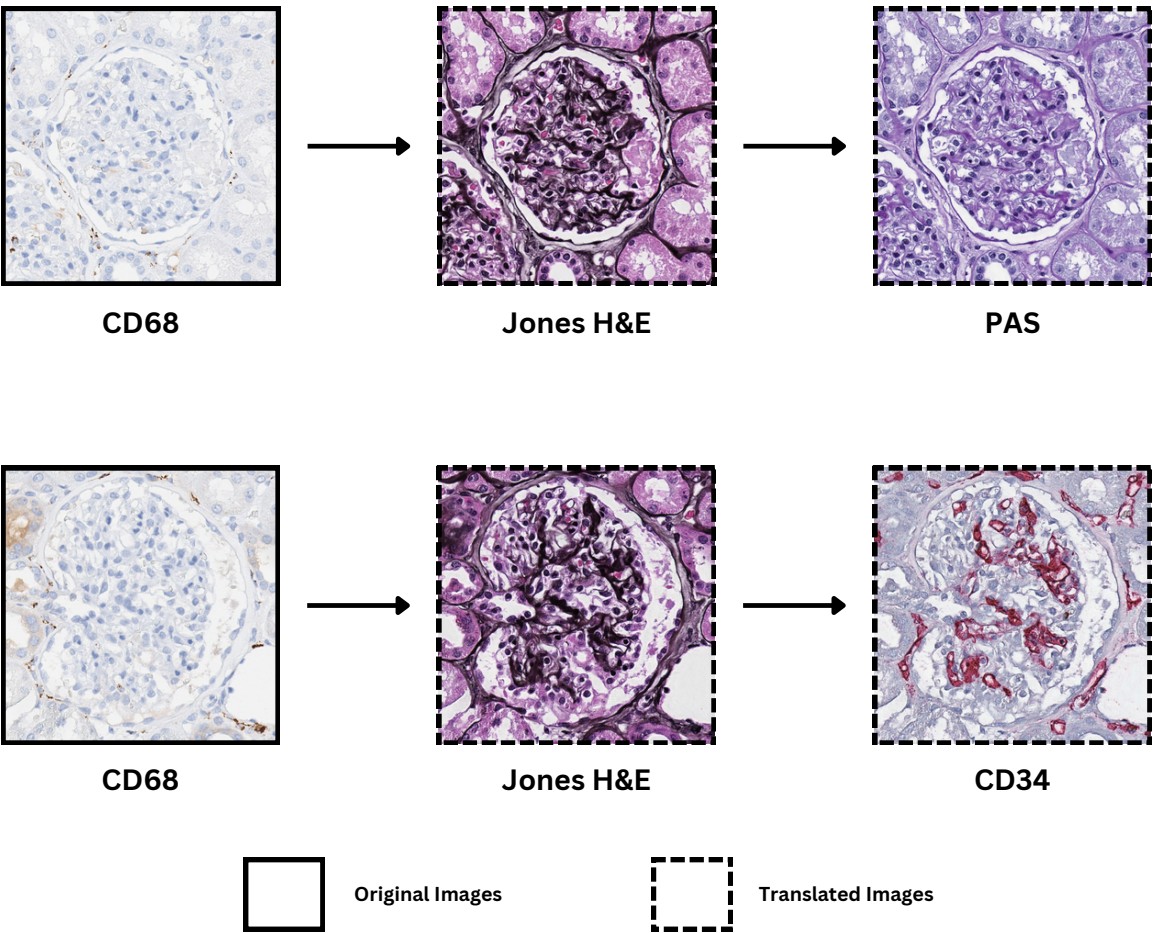

Figure 3:   StairwayToStain translation paths for CD68 → PAS and CD68 → CD34.

**PAS**                                     **CD34**

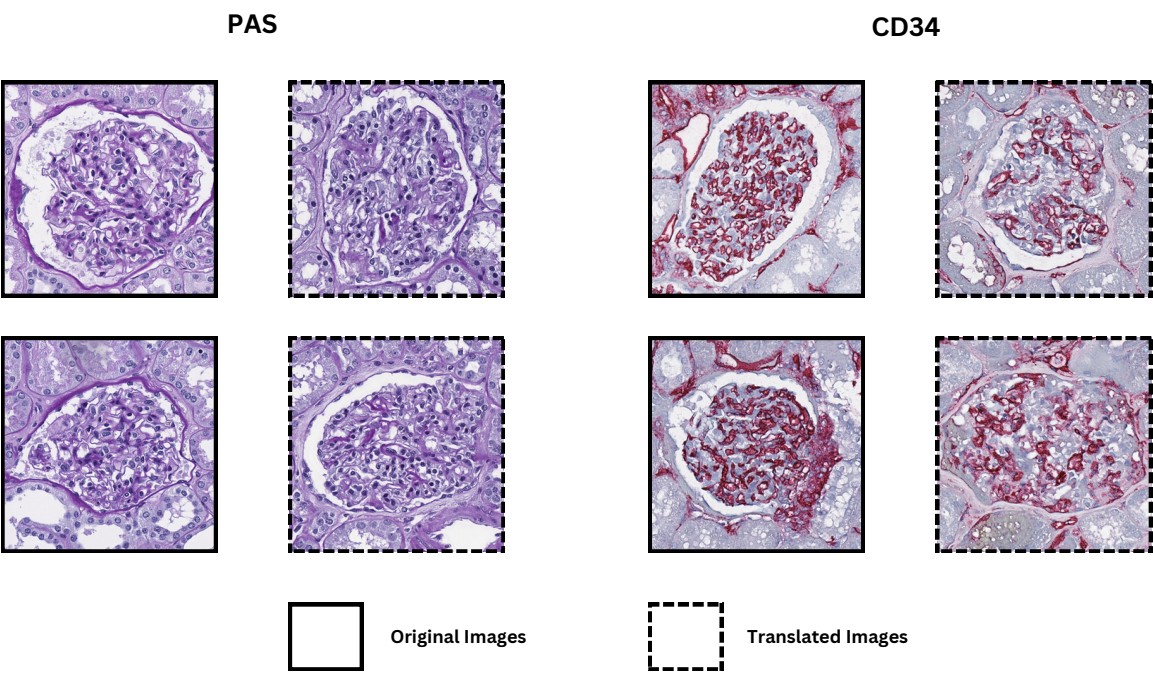

Original Images          Translated Images

Figure 4: Example of original patches and StairwayToStain translated patches for PAS and CD34.

