# OpenReview forum: "StairwayToStain: A Gradual Stain Translation Approach for Glomeruli Segmentation"
_MICCAI.org/2024/Workshop/COMPAYL — COMPAYL 2024_

### Official Review · Reviewer_7NGr · 2024-07-08
**Review of Submission 17**

**Custom Rating:** 4
**Confidence:** 5

**Review:**

The paper addresses the challenging issue of histopathology image translation with significant differences in staining domains. It employs a domain shift matrix quantified by the Wasserstein distance and uses Dijkstra's algorithm to find the optimal intermediate staining path, improving results through gradual color transition. This approach has been validated to be effective in challenging translation tasks, such as CD68→PAS and CD68→CD34.

The methodology of the paper is intuitive and effective, with good comprehensibility. However, there are aspects in the organization of the paper and the presentation of results that need improvement.

Insufficient presentation of results. The paper involves a large number of modality transformations and should provide sample images for each modality as well as the translation results achieved by CycleGAN and StarGAN. The effectiveness of the translation is evaluated through U-Net segmentation accuracy for glomeruli; visual results of the segmentations, especially for the cases with lower metrics in Table 3, should be provided.

To more fully demonstrate the effectiveness of the approach, there are actually 10 different translation tasks among the 5 staining types. It is suggested to supplement the complete results of these in future work (especially the difficult example of CD34→SR, as shown in Figure 2).

In the Introduction, many topics related to domain adaptation are mentioned, but this paper is essentially about image-to-image translation. The related work introduction could be more targeted. Additionally, the paper emphasizes the issue of artifacts when the translation fails but does not provide examples to support this claim.

Some other concepts could be further clarified, such as "information poor," "information rich," and "measured STS DSS." Additionally, it should be clarified whether DSM and DSS refer to the same concept.

---

### Official Review · Reviewer_7qYo · 2024-07-10
**Indirect stain translation through shortest path estimation**

**Custom Rating:** 2
**Confidence:** 4

**Review:**

Considering different stains at hand, the paper proposes to allow indirect translation from a source to a target stain through the possible introduction of intermediate stain(s) (e.g. H&E). The shortest translation path from one stain domain to another is estimated based on the domain shift metric and can travel through multiple intermediate stain domains. While the proposed method is elegant, some aspects of the methodology remain unclear and the reported results are significantly lower than the current state of the art.
-	It would be nice to explain why the graph is directional, more specifically why the distances reported by the in the Average Domain Shift score table are not symmetric. One could intuitively expect that the distance between CD68 and CD34 equals the distance between CD34 and CD68.
-	It is unclear how the downstream task of Glomeroli segmentation is evaluated, in particular which labeled dataset is used to generate the numbers reported in Table 3. It seems that the task is evaluated on the “CD68” labeled dataset and that the images from the source domain “CD68” are translated to “PAS” and “CD34” at test time and that the “PAS” and “CD34” specific models are applied on the so-translated CD68 images. If so, the method does not seem to leverage the availability of multiple labeled datasets for the different stains (Table 1). It could be relevant to compare the proposed approach against a train-time stain translation – i.e. translating all labeled images from each stain to CD68 and train a CD68 model on all the translated labeled images.
-	Reporting the SOTA results in Table 3 would facilitate the comparison of the proposed method with the HistoStarGAN. The proposed method yields very low recall vs. current state-of-the-art.
-	There is a typo in the table, assuming that the header should read CD34 – as in the text, and not CD39.

---

### Official Review · Reviewer_EZ3f · 2024-07-10
**Interesting method to address large domain shifts in stain transfer applications**

**Custom Rating:** 4
**Confidence:** 5

**Review:**

Authors have proposed a novel and intriguing approach to transferring histological images between different stains. This method effectively handles large domain shifts through a multi-step (gradual) stain transformation. The optimal stain transfer steps are determined by finding the shortest path between nodes in a graph, where the nodes represent different domains and the edges represent pre-calculated domain shift measures.

I find the proposed approach interesting, and the results, at least for one of the scenarios, are promising. Therefore, I believe this work warrants more attention and in-depth investigation. To enhance the manuscript, consider addressing the following questions:

_Major Comments:_

**M1 - Additional Stain Transfers**: Have the authors considered performing stain transfers between other types of stains (other paths beyond CD68 -> CD39/PAS)? If so, it would be beneficial to report the results of these experiments, perhaps in the supplementary material section.

**M2- Usual Selected Path**: What is the typical path selected for stain transfer? This approach can provide insights into whether there is a consistently optimal path, such as always transferring from Stain1 -> H&E -> Stain2, with H&E as an intermediate stain, as it worked best in the reported cases.

**M3 - UNet Model Pretraining**: It is crucial to mention how the segmentor UNet model was pretrained, as this model seems to play a significant role in the results discussed in the manuscript.

_Minor Comments:_

**m4 - Multi-domain Supervised 1 Method**: What is the Multi-domain Supervised 1 method mentioned on page 2? It is introduced without any citation or explanation.

**m5 - Domain Definitions**: What exactly are information-poor and information-rich domains? What criteria are used to define a domain as either of these?

**m6 - HistoStarGan Results**: Why not add the HistoStarGan results to Table 3?

**m7 - Combining Sections**: Consider combining the results and discussion sections into a single "Results and Discussion" section. This will provide more writing space and flexibility in reporting and discussing the results.

**m8 - Incorporating Figures**: Try fitting one of the qualitative results figures (such as Figure 3) into the main manuscript. This can be done by reducing the size of Figures 2 and 3. Additionally, you can summarize Table 2 and reduce the text in Section 2.1 (since it is already published) if more space is needed.

---

### Decision · Program_Chairs · 2024-07-16

Accept